# Occurrence and diversity of *Campylobacter* species in diarrheic children and their exposure environments in Ethiopia

Dinaol Belina[1,2]*, Tesfaye Gobena[3], Ameha Kebede[2], Meseret Chimdessa[2], Bahar Mummed[1], Cecilie Amalie Neijendam Thystrup[4], Tine Hald[4]

1 College of Veterinary Medicine, Haramaya University, Dire Dawa, Ethiopia, 2 School of Biological Sciences and Biotechnology, Haramaya University, Dire Dawa, Ethiopia, 3 College of Health and Medical Sciences, Haramaya University, Harar, Ethiopia, 4 National Food Institute, Technical University of Denmark, Lyngby, Denmark

* belina.dinaol@gmail.com

## Abstract

*Campylobacter* is a major zoonotic foodborne pathogen that poses a significant public health threat, particularly among children and immunocompromised individuals. However, data on the occurrence and sources of *Campylobacter* infection remain scarce in Ethiopia. This study assessed the occurrence, diversity, and relationships between *Campylobacter* from diarrheic children and potential exposure sources using whole-genome sequencing. Through case-based tracing, animal, food, and environmental samples were collected from Harar town and Kersa district between November 2021 and January 2023. *Campylobacter* was identified using selective media, and DNA was extracted and sequenced with the Illumina NextSeq 550 instrument. Sequence reads were analysed using bioinformatics tools. The overall *Campylobacter* prevalence in the exposure sources was 5.5%, with 6.0% in urban and 5.0% in rural settings. *Campylobacter* detection was 1.8 times more likely in household samples (8.7%; OR = 1.8; 95% CI: 0.7–4.5) than in samples from marketplaces. The occurrence of *Campylobacter* in food was 4.2%, with no significant differences across the meat, milk, and other food categories. The likelihood of *Campylobacter* contamination in the environment was 5.8 times higher in the presence of poultry (17.7%; OR = 5.8; CI: 1.1–30.6) compared to shoats. Sequence analysis identified a low *Campylobacter* spp. diversity comprising only *C. jejuni* and *C. coli*, which were characterized by 8 distinct sequence types (STs). Phylogenetically, the majority of the sequenced case isolates were clustered with isolates from either caretakers, environmental exposures, or both. In conclusion, *Campylobacter* was detected in various exposure sources of diarrheic children, and its occurrence did not differ significantly between Kersa and Harar or among food items. The majority of isolates shared MLST profiles and clustered together, demonstrating the involvement of multiple vectors in the transmission of the pathogen. Genome-based integrated studies supported by an attribution model are recommended to determine the relative contribution of each source.

**Data Availability Statement:** The summary of the data used in this study are presented in the paper, and the raw sequences has been submitted to the European Nucleotide Archive (ENA) under accession number PRJEB73590.

**Funding:** The study is part of FOCAL (Foodborne Disease Epidemiology, Surveillance, and Control in African Low- and Middle-Income Countries). FOCAL is a multi-partner, multi-country research project co-funded by the Bill & Melinda Gates Foundation and the Foreign, Commonwealth, and Development Office (FCDO) of the United Kingdom Government [grant agreement investment ID OPP1195617] awarded to TH. However, the funders did not have a role in the study design, data collection and analysis, or the decision to prepare or publish the manuscript.

**Competing interests:** The authors have declared that no competing interests exist.

## Introduction

*Campylobacter* is a bacterium that commonly inhabits the gastrointestinal tract of most warm-blooded animals. It is one of the most important zoonotic foodborne pathogen and poses a significant public health threat globally. Diarrheagenic foodborne pathogens (FBPs) are any biological agents that have the potential to induce infectious or toxic illness-associated diarrhea in humans upon ingestion of contaminated food. In recent years, *Campylobacter* has been identified as one of the four key global causes of diarrheal diseases, with a greater burden on developing countries and children under the age of five (UFC) [1–3].

*Campylobacter* is transmitted primarily through the fecal–oral route, which occurs when individuals ingest contaminated food or water, as well as through direct contact with infected animals or their feces. In Ethiopia, the incidence rate of diarrheal episodes in children under five years of age is 186.7 thousand per year, and it is the second leading cause of death among this age group, killing 150.7 children per 100,000 people annually [4]. *Campylobacter* spp. have been reported in diarrheic children as important causes of diarrhea, with an occurrence rate of up to 20% [5], and *C. jejuni* is the predominant species [5–7].

In Ethiopia, where livestock farming and subsistence agriculture are prevalent and food safety measures are poor, the occurrence of *Campylobacter*-associated diarrhea can be influenced by multiple risk factors. These include close proximity to carrier animals, particularly chickens, inadequate access to safe drinking water, socioeconomic status, and limited knowledge about proper hygiene practices. The habit of consuming raw or undercooked meat and unpasteurized dairy products can also increase the risk of campylobacteriosis [7–10].

Previous studies in Ethiopia have also highlighted a high prevalence of *Campylobacter* in animals, food or water, with rates reaching 56.5% in animals [11], 13% in poultry, 10.5% in water [12], and 9.3% in animal-sourced foods [13]. In eastern Ethiopia, studies have identified limited awareness of zoonotic diseases and poor sanitation, including open defecation practices in some areas [14–16]. Consequently, the occurrence of *Campylobacter*, like other diarrheagenic pathogens, may be more prevalent in such settings, which may be further aggravated in rural areas [17]. Moreover, *Campylobacter*-associated diarrhea may be more severe in young children because of their developing immune systems and exploratory behaviors.

However, previous studies have reported the prevalence of *Campylobacter* in humans or animals separately using routine culture-based methods, with a limitation of studies on tracing the source of *Campylobacter* infection using advanced diagnostic tools in Ethiopia. Therefore, a comprehensive study on the occurrence of *Campylobacter* at the human-animal-environment interface using genomic analysis is important for designing targeted interventions and effective integrated prevention and control strategies for *Campylobacter* infection. This study aimed to assess the occurrence, diversity, and relationships between *Campylobacter* isolated from diarrheic children and their potential source exposures, including human, animal, food, and environmental samples in Harar town and Kersa district, using a whole-genome sequencing-based phylogenetic analysis.

## Materials and methods

### Study area

The study was conducted in Harar town and Kersa district, which are located in the eastern part of Ethiopia. Harar is the capital city of the Harari People National Regional State and is one of the most densely populated and oldest cities in the country. The Harari region is divided into six urban and three rural administrative districts, with Harar being the only

urban area in the region. The total population of the Harari region is estimated at 257,000, with 11% of its population aged between 0 and 4 years [15]. Unlike other regional states in Ethiopia, over 54.2% of the population of the Harari region lives in urban area, in Harar city. The region is served by 7 hospitals, 8 public health centers, 32 health posts, and 25 clinics [18].

On the other hand, Kersa district is predominantly rural, with three small towns (Kersa, Weter and Langhe) and 35 rural subdistricts. The district has a total population of 172,626, with most inhabitants being farmers. The Kersa district is located approximately 44 km west of Harar town and has 6 health centers and 28 health posts but no hospitals [18,19].

Both Harar town and Kersa district are covered by the Health and Demographic Surveillance Sites (HDSS) established by Haramaya University. The HDSS in these areas has provided a GPS-based ID for each household under the surveillance follow-up, which allows better tracing of households.

## Sample types and collection procedures

From November 2021 to January 2023, a case-based source exposure tracing study was conducted in Harar town and Kersa district. The cases were children under the age of five (UFC) with diarrhea visiting local healthcare facilities. The study included various exposure sources that were reported by caretakers as direct or indirect contact of the children shortly before the onset of diarrheal symptoms and signs. Through this case-based tracing, various environmental samples were collected from the immediate physical environments of known diarrheic UFC (**Fig 1**).

In this study, the term "exposure source" indicates any accessible sample sources that were identified by caretakers as having been ingested by or come into contact with children prior to the onset of diarrhea. These include food, animal, wastewater, and water sources. The "direct exposure environment" is the immediate physical environment where diarrheic children have direct contact, such as their households. On the other hand, the "indirect exposure environment" referred to the physical locations where the family members, specifically the caretakers, of the UFC had contact and brought items used by the children (e.g., local market).

Before the exposure sources were traced, children with diarrhea visiting Hiwot Fana specialized University Hospital, Kersa, and Adelle health centers were screened for eligibility. The families of the children were required to be residents of either Harar town or the Kersa district. The details of the case (UFC) inclusion criteria are described in our recently published article [18].

Following case identification, face-to-face interviews were conducted with caretakers to obtain relevant background information and recent exposure details. These details included a history of what the child has ingested, the origin of the consumed food (self-produced, purchased from a market, etc.), information about animal contact, specific household addresses, and sources of drinking water used in the household prior to developing diarrhea. Caretakers were also asked about possible exposure to irrigation water, recreational water, or wastewater linked directly or indirectly to households (**Fig 1**).

**Collection of exposure source samples.** After the interviews with caretakers and with their consent, exposure source samples were collected by visiting households and other identified contact sites within 1–2 days of collecting diarrheal stool samples. The questionnaire responses were used as guidance for the sample collection, which were initially completed on a tablet using Epi Info 7 software at the health facilities, enabling the identification of specific locations and types of samples to be collected prior to the visits.

During the household visits, food samples, including stored food and food leftovers (when available), were collected. Drinking water used in households and any wastewater linked to the

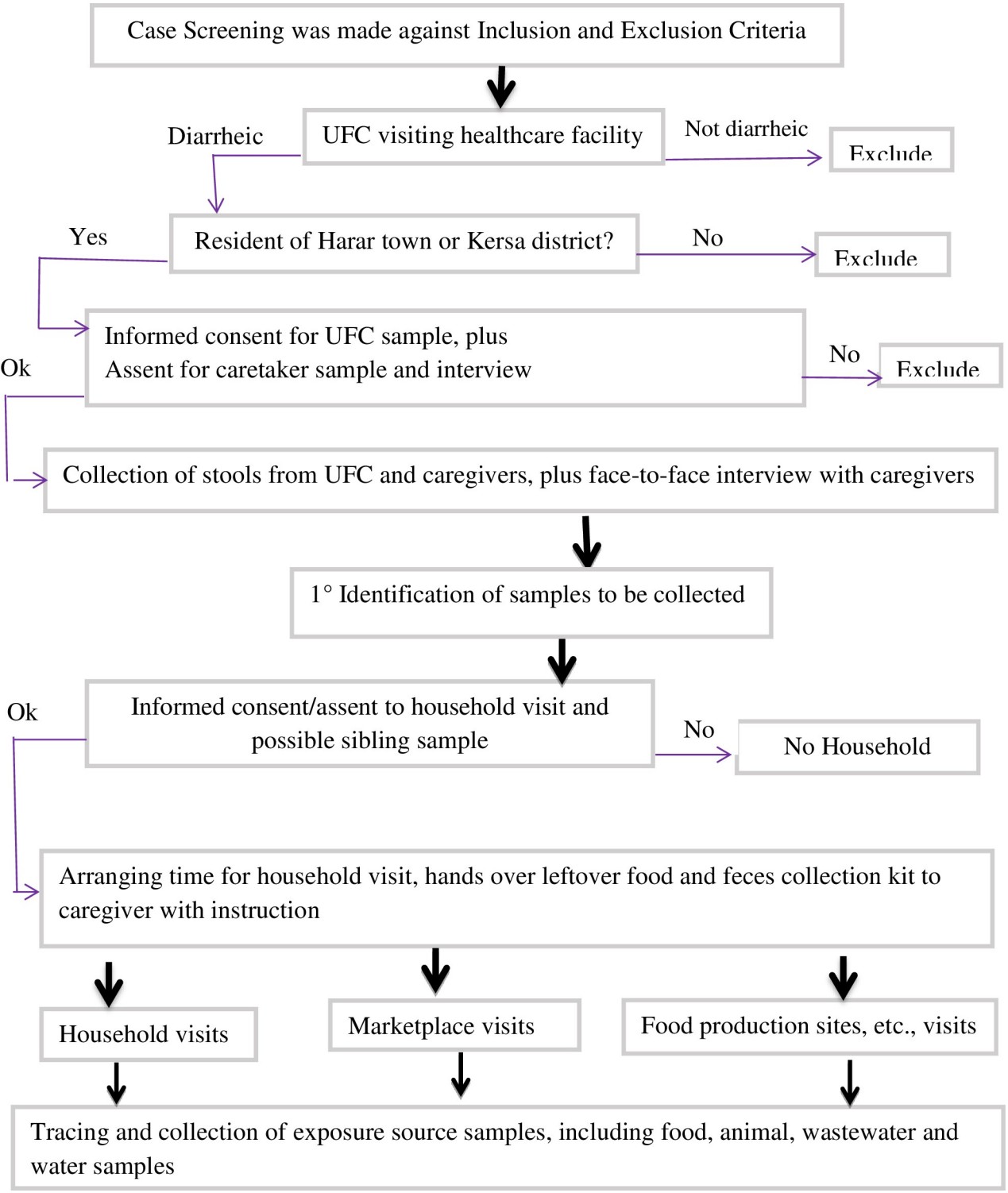

**Fig 1. Procedures used for case identification and collection of samples from exposure environments.**

households were also collected. Additionally, wastewater and animal feces were aseptically collected from households. Furthermore, exposure source samples were collected from slaughterhouses, marketplaces, hotels and restaurants, grocery stores, butcher shops, nearby farms and

lakes where the children had recent direct or indirect contact before becoming sick. The details of exposure source tracing is described in our paper Gobena *et al* [20].

## Sample collection and transportation

All samples were collected aseptically into sterile stomacher bags (Fisher Scientific Biotech line) from the identified sites, labeled, and immediately placed in a cold box with ice packs. The samples were transported on the same day to the Microbiology Laboratory of the School of Medical Laboratory, Haramaya University.

The sample preparation adhered to a specific standard operating procedure (SOP), with minor modifications to the previously described ISO 10272–1 protocol [21,22]. The samples were prepared in Bolton broth at a 1:9 (v/v) sample-to-Bolton broth ratio, and each sample was gently mixed before the analytical volume was determined. For animal feces, approximately 5 g of feces was gently mixed using a sterile spatula, and 1 g of the well-mixed feces was transferred into a sterile Falcon tube containing 9 mL of Bolton broth, which was then microaerophilically incubated. In the case of wastewater samples, 25 mL of each sample was prepared in 225 mL of Bolton broth. Similarly, a 25-to-225 sample-to-broth proportion was used for food sample preparation, whereas 125 mL of each water sample was prepared in an equal volume of Bolton broth. However, when sufficient volumes of food were not available, 10 g of well-mixed food samples were suspended in 90 mL of Bolton broth.

The prepared samples were incubated microaerophilically at 37˚C for 4 hours, followed by incubation at 42˚C for 40 ± 4 hours. A loop full of suspension from this selective enrichment was subsequently plated on blood-free *Campylobacter* selective agar, the modified charcoal cefoperazone deoxycholate agar (mCCDA). The isolation and identification of *Campylobacter* was conducted on the basis of colony morphology on mCCDA plates and microscopic appearance, as well as oxidase and catalase test results, as previously described [5,18,23,24]. The pure colonies of *Campylobacter* isolates obtained from the samples were preserved at -80˚C until they were used for DNA extraction.

## DNA extraction and whole-genome sequence of *Campylobacter*

The *Campylobacter* isolates preserved at -80˚C were revived by subculturing on brain heart infusion agar, and DNA extraction was performed using the Quick-DNA Fungal/Bacterial Miniprep Kit following the manufacturer's instructions [25]. The quality and quantity of genomic DNA was confirmed using a Qubit 2 fluorometer. Only samples with an absorbance A260/A280 ratio of 1.8 or above were considered for further analysis. The DNA samples meeting the quality criteria were shipped on ice to Admera Health Biopharma (San Diego, USA) through GenoHub shipping services for whole-genome sequencing (WGS). At Admera Health Biopharma, the DNA samples were prepared using a Nextera XT Library Preparation Kit for Illumina sequencing, according to the manufacturer's instructions [26]. The samples were then sequenced using an Illumina NextSeq 550 instrument, generating 2x150 bp paired-end reads.

## Data quality control

Throughout the study, data quality was maintained by adhering to standard protocol. Quality controls for the bacteriological analysis were performed as described previously [18]. The concentration and purity of the DNA were assessed spectrophotometrically using a NanoPhotometer (Mettler-Toledo, Singapore). The paired-end reads were preprocessed using an in-house QC and assembly pipeline using BBDuk [27] and FastQC [28] for adapter

trimming and quality checking. Reads from the WGS isolates were assembled using SPAdes [29] (version 3.9).

## Data management and analysis

**Statistical analysis.**   The laboratory data were entered into Microsoft Excel, cross-checked, and exported into SPSS version 22 for analysis. Descriptive statistics were used to calculate and interpret the study variables. To determine the prevalence of *Campylobacter*, the total number of positive samples was divided by the total number of samples processed in the laboratory, which was multiplied by 100. The Pearson chi-square ($X^2$) test was used to assess the associations between explanatory variables, and observed differences were considered statistically significant when $P \leq 0.05$. The monthly *Campylobacter* occurrence rate was calculated by considering the number of *Campylobacter* isolates obtained in a given month and the total number of samples analysed in the laboratory in that particular month. Moreover, a binary logistic regression model was executed, and the odds ratio was employed to measure the associations between independent variables and the occurrence of *Campylobacter*.

**Bioinformatics analysis.**   The *SpeciesFinder* and *KmerFinder* tools, accessed from the Center for Genomic Epidemiology (CGE) page (https://www.genomicepidemiology.org/services/) were for the determination of species of *Campylobacter* isolates. For isolates identified as *Campylobacter*, seven-locus multilocus sequence typing (7gMLST) was conducted on the sequences using the pipeline developed by Larsen et al [30] (https://cge.food.dtu.dk/services/MLST/), and the schemes developed for *Campylobacter* spp.[31].

**Phylogenetic tree.**   The phylogenetic tree was constructed to determine the relationships between the WGS isolates using the Call SNPs and Infer Phylogeny (CSI) pipeline [32], which uses single nucleotide polymorphisms (SNPs) to find sequence variations shared between the samples. The tool *FastTree* was used to construct maximum likelihood phylogenetic clusters on the basis of these sequence variations in the genome shared between the isolates. To visualize and annotate the phylogenetic trees, iTOL was used [33]. Clusters were identified as isolates with SNP differences less than 30 (SNP < 30). The raw sequences of isolates analysed in this study have been submitted to the European Nucleotide Archive (ENA) under accession number PRJEB73590.

## Ethical considerations and participants' consent

The study was part of a FOCAL project that was reviewed and approved by the Institutional Health Research Review Committee of Haramaya University and the National Research Ethics Committee (Ref Number: MoSHE/RD/14.1/9849/20). A formal letter of support was subsequently obtained from Haramaya University to communicate with the Harari region, the East Hararghe and Kersa district health bureaus, and the involved local healthcare facilities.

The caregivers were fully informed about the study and their right to continue or withdraw from it at any time. Samples were collected only after receiving informed consent from each child's caretaker in the presence of healthcare facility staff members. Upon the caregivers' consent, household visits and the collection of possible contact samples were made. The confidentiality of the participants' information was strictly maintained by coding personal/household data. For disease management and suggested interventions, laboratory results were disclosed to the relevant stakeholders by organizing national and international official meetings, where stakeholder representatives were invited and participated

**Table 1. Occurrence of *Campylobacter* in various sources of environmental samples in Kersa and Harar, Ethiopia (N = 438).**

| Samples Source and Type | | N. Analyzed | Positive | OR (95% CI) | | P value |
|---|---|---|---|---|---|---|
| **Source** | Abattoir | 132 | 5(3.79) | 0.73(0.25–2.14) | | 0.56 |
| | Household | 92 | 8(8.7) | 1.8(0.68–4.52) | | 0.24 |
| | Market | 214 | 11(5.14) | 1 | | – |
| **Type** | Animal | 164 | 11(6.71) | 0.72(0.25–2.03) | | 0.53 |
| | Food | 168 | 7(4.17) | 0.44(0.14–1.35) | | 0.15 |
| | Wastewater | 66 | 6(9.09) | 1 | | – |
| | Water sources | 40 | 0(0) | – | | – |
| **Study area** | Harar town | 217 | 13(5.99) | 1.3(0.53–2.78) | | 0.64 |
| | Kersa district | 221 | 11(4.98) | 1 | | – |
| | **Total** | **438** | **24(5.48)** | – | – | |

## Results

### *Campylobacter* prevalence and monthly occurrence

The overall prevalence of *Campylobacter* in the exposure environments of diarrheic children was 5.5% (24/438), with 6.0% and 5.0% in urban and rural settings, respectively. The prevalence of *Campylobacter* was 6.7% in animal feces, 4.2% in food, and 9.1% in wastewater samples. The detection rate of *Campylobacter* in samples collected from households was 1.8 times higher (OR = 1.8; 95% CI: 0.68–4.5) than in samples collected from marketplaces. Nevertheless, there was no significant difference (p > 0.05) in the occurrence of *Campylobacter* among the sample types, sample sources, or between urban and rural study sites (**Table 1**).

### Water sources–drinking, recreational and irrigation water

The contamination rate of food with *Campylobacter* was 4.2%, with 3.2% in meat, 5.3% in milk, and 5.4% in other food categories. None of the cooked food samples tested positive for *Campylobacter*. However, there were no significant differences in the distribution of *Campylobacter* among food types and sources (p > 0.05). Logistic regression analysis revealed that the likelihood of *Campylobacter* contamination in the environment was 5.8 times higher in the presence of poultry feces (17.7%; OR = 5.8; CI: 1.1–30.6) compared to shoat feces. The occurrence of *Campylobacter* in animal wastewater was 5.4%, whereas its correspondence rate in mixed waste from both animals and humans was 13.8% (**Table 2**).

This study revealed fluctuations in the monthly occurrence patterns of *Campylobacter* throughout the study period, which spanned from November 2021 to January 2023, and identified a peak of occurrence in February 2022. The second highest monthly occurrence was observed during the rainy season, with a peak of 6.7% in July 2022 (**Fig 2**).

### Whole-genome sequence (WGS) analysis

Among the 51 *Campylobacter* isolates identified from children with diarrhea, their caretakers, and potential source exposures, 37 isolates were subjected to WGS as they had good-quality DNA for library preparation. The analysis revealed that only *C. jejuni* and *C. coli* were detected, with frequency rates of 62.16% and 37.84%, respectively. Among the 13 sequenced case isolates, the majority (76.9%) were identified as *C. jejuni*. The distribution of *Campylobacter* spp. across sample types in eastern Ethiopia is presented in **Table 3**.

This study identified 8 distinct sequence type (ST) profiles for the 37 sequenced *Campylobacter* isolates using MLST analysis. Nineteen (19) isolates were assigned specific ST profiles,

**Table 2. Descriptive statistics of *Campylobacter* by animal species, food item and wastewater samples traced as potential contacts of diarrheic children.**

| Sample type and source | | N. analysed | Positive (%) | OR (95% CI) | P value |
|---|---|---|---|---|---|
| **Food** | Meat | 93 | 3(3.23) | 0.59 (0.12–3.02) | 0.79 |
| | Milk | 19 | 1(5.26) | 0.98(0.10–10.04) | |
| | Other food | 56 | 3(5.36) | 1 | |
| | Abattoir-linked | 52 | 1(1.92) | 0.76(0.07–8.54) | |
| | Household-linked | 37 | 4(10.81) | 4.67(0.81–26.74) | 0.07 |
| | Market-linked | 79 | 2(2.53) | 1 | |
| | Cooked | 27 | 0 | – | |
| | Raw | 141 | 7(4.96) | – | 0.24 |
| | Animal sourced food (ASF) | 119 | 4(3.36) | 0.53(0.15–2.48) | |
| | Other | 49 | 3(6.12) | 1 | 0.42 |
| **Animal feces** | Cattle and Camel | 74 | 3(4.05) | 1.14(0.18–7.07) | |
| | poultry | 34 | 6(17.65) | 5.79(1.10–30.56) | 0.02 |
| | Shoat | 56 | 2(3.57) | 1 | |
| | From abattoirs | 37 | 2(5.41) | 0.73(0.14–3.67) | |
| | From households | 31 | 2(6.45) | 0.88(0.17–4.46) | 0.92 |
| | From marketplaces | 96 | 7(7.29) | 1 | |
| **Waste-water** | From animals | 37 | 2(5.41) | 0.36(0.06–2.11) | 0.25 |
| | Mixed waste | 29 | 4(13.79) | 1 | |

Mixed waste: Mixed wastewater from both humans and animals.

whereas the remaining *Campylobacter* isolates could not be assigned STs and had previously unknown STs. The analysis revealed that the most abundant ST profile was ST353, followed by ST19 and ST1365. MLST analysis also revealed that 84.6% (11/13) case isolates shared ST profiles with at least one isolate from caretakers or environmental exposures (**Fig 3**).

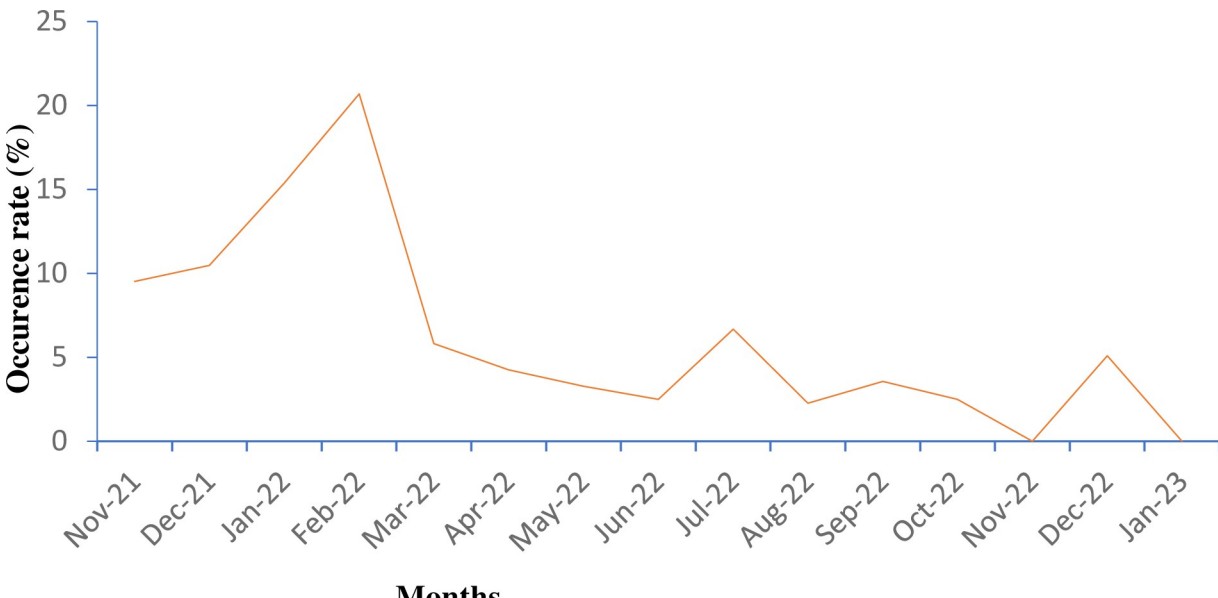

**Fig 2. Monthly occurrence patterns of *Campylobacter*.** Fluctuations in the monthly occurrence patterns of *Campylobacter* in the exposure environments of children with diarrhea between November 2021 and January 2023 in east Ethiopia.

**Table 3. Frequency distributions of *Campylobacter* spp. detected in children with diarrhea and their potential exposure environments in eastern Ethiopia (n = 37).**

| Sample type | Frequency (%) | | Total |
|---|---|---|---|
| | *C. jejuni* | *C. coli* | |
| Environmental | 10(50) | 10(50) | 20(54.05) |
| Caretaker | 3(75) | 1(25) | 4(10.81) |
| Diarrheic UFC | 10(76.92) | 3(23.08) | 13(35.14) |
| **Total** | **23(62.16)** | **14(37.84)** | **37(100)** |

The maximum likelihood phylogenetic analysis revealed that *Campylobacter* isolates formed multiple small clusters (clades), irrespective of their sources, with minimum variation among isolates within each cluster (SNP < 30). The phylogeny suggested that there was transmission of *Campylobacter* among humans (caretakers), animals, food products, and wastewater from different sites in the vicinity that were either directly or indirectly contacted or accessed by children in the study area.

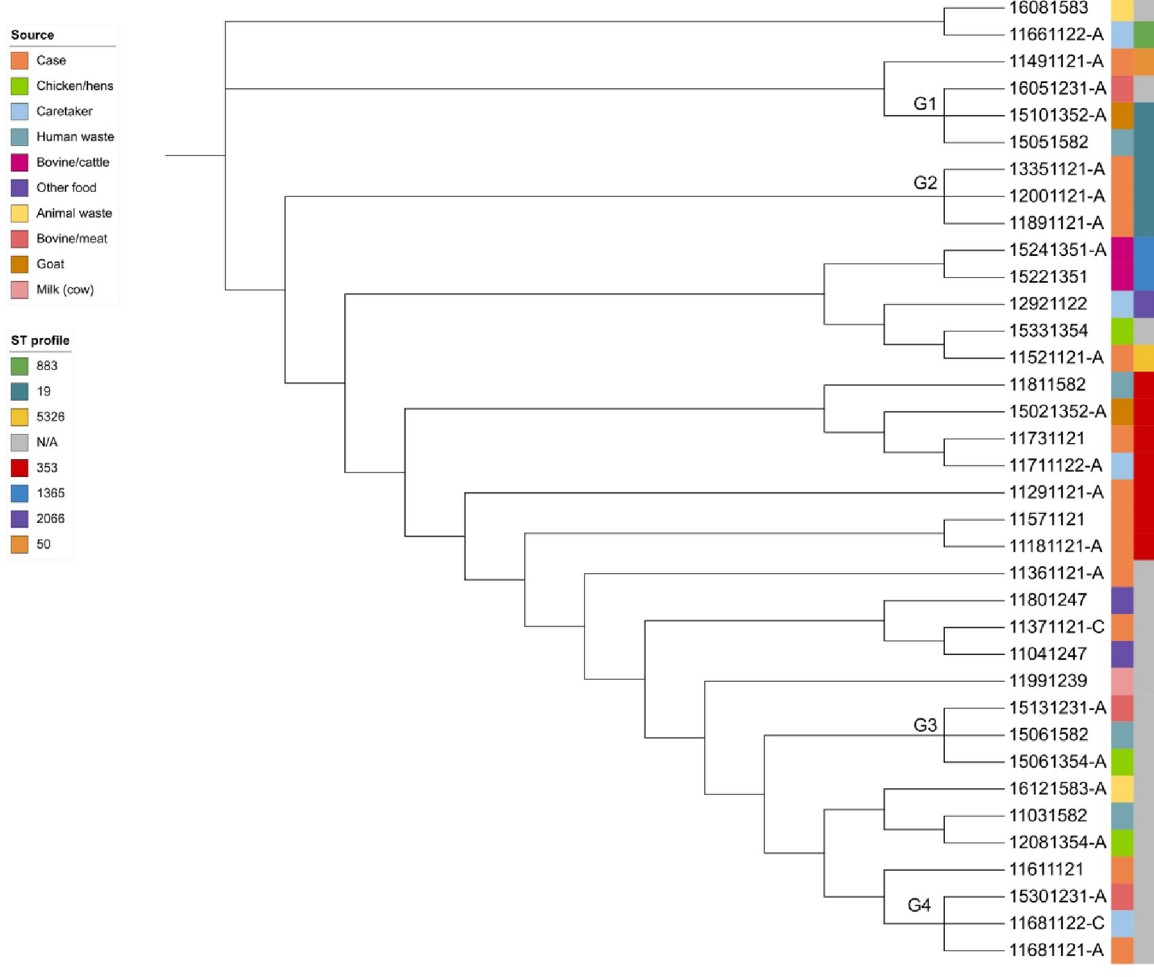

**Fig 3. Phylogenetic tree showing *Campylobacter* isolates from various sources.** The labels are colored according to the isolation source and sequence type (ST) profiles following the color scheme. Four clades with three samples in each were identified on the basis of SNP differences (SNP < 10) and are denoted G1- G4 in the tree.

Notably, the isolates in cluster G4 originated from a child with diarrhea, a caretaker in the same household, and a bovine meat product from a local market nearby, demonstrating transmission of the pathogen between the household and the food product. In the other clusters, the case isolates were exclusively grouped together in a few clades. For example, clade G2 contained isolates from three diarrheic children admitted to a local healthcare facility, suggesting that the children contracted closely related *Campylobacter* strains that could have originated primarily from a common source in the locality before their illness.

Another clade, G3, showed a close similarity among *Campylobacter* isolates from chicken, human waste collected from the same local market, and meat from another nearby market. The isolates within clade G1 were identified from abattoir-linked meat, goat feces from the same town, and restaurant-linked wastewater, indicating that some *Campylobacter* strains had spread across different markets in the study area, with the potential to contaminate various food and environmental samples. Notably, *Campylobacter* isolates within each clade denoted as G1-G4 were genetically closely related (SNP < 10).

## Discussions

*Campylobacter* poses a significant public health risk and is commonly acquired through the consumption of contaminated food, contact with infected animals, and exposure to contaminated environmental samples [5,34]. Assessing Campylobacter occurrence and understanding the genetic relationships between case isolates and isolates from various sources is important for tracing the sources of *Campylobacter* infection. The present study identified *Campylobacter* across all traced sample types, with the exception of water sources. The overall *Campylobacter* occurrence rate of 5.5% in the physical exposure of diarrheic children in this study is comparable with the PCR results of a study conducted in Egypt [35]. However, the prevalence is lower than the recently reported 7.7% occurrence in different environmental samples [36] and 21% in diarrheic children and animals [5] in Ethiopia. Moreover, pooled prevalences of 10.2% [13] and 14.9% [37] for *Campylobacter* in the environment were reported in meta-analyses conducted in Ethiopia and Iran, respectively.

The occurrence of *Campylobacter* was not significantly different (P > 0.05) between Harar town and Kersa district, highlighting its importance in both the urban and rural settings of the current study area. Similarly, a study conducted in Ethiopia by Yirgalem et al [36] reported no statistical differences among the study sites. Other studies have reported that diarrheagenic pathogens remain a significant concern in resource-limited countries such as Ethiopia [6,38], indicating poor environmental sanitation regardless of the setting. However, the health impacts of FBPs, including *Campylobacter*, are further aggravated in rural areas [17]. Nevertheless, there have been no prior studies on the occurrence of *Campylobacter* in the physical exposures of diarrheic children in urban and rural settings in Ethiopia that could be used for comparison with our current findings.

Food samples accounted for 4.2% of the occurrence of *Campylobacter* in the environments of children with diarrhea (4.17%, OR = 0.44; 95% CI: 0.14–1.35). The prevalence was 3.4% in animal-sourced food (ASF) and 6.1% in other food categories. However, a previous study reported a higher prevalence of *Campylobacter* (9.3%) in food in Ethiopia [13]. The occurrence of *Campylobacter* in meat in this study was lower than the rates recently reported for meat in Ethiopia (6.8%) [36], beef offals from Queensland (10%), New South Wales (21%) [39] and goat meat in Africa (29.7%) [40]. Data from many European countries have shown a 12.9% occurrence rate of *Campylobacter* in ready-to-eat meat and a 79.7% occurrence rate in non-ready-to-eat meat and meat products [41].

The majority of the meat samples analysed in this study were collected at the abattoir level or upon arrival at butcher shops or households. The detection of *Campylobacter* in food linked

to abattoir (particularly meat) samples suggests contamination of meat by animal waste or abattoir personnel. This cross-contamination is further evidenced by the identification of *Campylobacter* in slaughterhouse wastes. However, the relatively higher rate of *Campylobacter* in food samples collected from households could be attributed to contamination with poultry feces [42], as there were typically more poultry at home than at other sample collection sites included in this study.

Among the traced milk samples, 5.3% tested positive for *Campylobacter*. This finding is comparable to the prevalence rate of 5.4% documented in nonready-to-eat milk and milk products [41]. However, the present *Campylobacter* prevalence in milk was lower than the rates reported in previous studies conducted in Ethiopia, which were 11% [9] and 20% [43]. A study analysing data from many European countries also revealed a greater occurrence of *Campylobacter* (24.2%) in ready-to-eat milk and milk products [41]. The consumption of raw milk/meat and meat/milk products is common in Ethiopia, and the current occurrence of *Campylobacter* in ASF (milk and meat) indicates the potential of ASF to transmit or spread *Campylobacter*. This also suggests the need to implement effective food safety measures, such as proper handling, cooking, and sanitation practices, to reduce the risk of *Campylobacter* or other FBP-associated infections.

In addition to milk or meat, *Campylobacter* was detected in other food items—RTE foods, fruits, vegetables, juices, and leftover injera mixed with wot—at a rate of 5.4%. In contrast, a lower occurrence rate of 0.3% for *Campylobacter* was reported in juices and other nonalcoholic beverages from Ethiopia [44]. Apart from Ethiopia, prevalence rates of 0.53% [45] and 36.6% [41] have been reported for *Campylobacter* in vegetables, fruits, and fresh produce. This study revealed that fruits and vegetables as well as RTE food purchased in the current study area have contributed to the occurrence of human campylobacteriosis. Importantly, the contamination of fresh fruits and vegetables with *Campylobacter* can occur through various means, including contact with contaminated water or soil or during handling and transportation. The current study further indicates that hygienic practices related to preparation and handling may be inadequate and require attention.

The absence of notable statistical variations in the distribution of *Campylobacter* among meat, milk, and other food categories (p > 0.05) demonstrates that the risk of contracting campylobacteriosis remains consistent regardless of the type of food consumed. On the other hand, the variations in *Campylobacter* prevalence among the studies may be attributable to differences in epidemiological factors and the level of food safety practices in different regions. Discrepancies in food types as well as nonuniform protocols for sampling and identification could have resulted in heterogeneity among studies, although some high-prevalence findings could be factual owing to an extensive variety of raw and RTE foods [46].

As shown in **Table 2**, there were no significant differences (p ≤ 0.05) in the occurrence of *Campylobacter* among animal feces collected from abattoirs, households, and marketplaces. This finding contradicts a previous study conducted by Chala et al [12], which reported a significant association between *Campylobacter* occurrence and animal feces from households. The overall prevalence of *Campylobacter* in the animals in this study (6.7%) is higher than a rate reported in a similar study (4.5%) from Bahir Dar [5] but lower than that reported in other previous studies, 56.5% [11] from Jimma, and 14.6% in a meta-analysis study in Ethiopia [13]. Furthermore, a higher prevalence of 13.5% at the farm level and 25.3% at the abattoir level has been reported in animal feces from other regions [37].

The occurrence of *Campylobacter* was significantly higher in poultry (17.65%; P = 0.04) than in cattle and camels, and shoats, which is consistent with the description that birds are the main reservoirs of *Campylobacter*, probably because of their elevated body temperature [47]. However, the prevalence in poultry in this study was lower than the 44% reported in a

similar study conducted in Ethiopia [5]. The prevalence of *Campylobacter* in cattle or camel feces in this study is comparable to that reported in a study conducted in Iran, which reported a prevalence of 5.3% in cattle and 4% in camels [48]. Similarly, a previous study in Ethiopia reported a higher prevalence of *Campylobacter* in poultry (86.6%) compared to cattle (48%), sheep (39%), and goats (33.3%) [11]. Conversely, another study documented higher *Campylobacter* prevalence in cattle (18.5%) and sheep (13.3%) feces than in poultry feces (13%) [12].

The variation in the prevalence rates among different studies may be attributed to differences in sample size, sampling protocol, and the inclusion of a varying number of animal species. In this study, the pooling of samples from the same species at the same site or in the same households may have affected the prevalence rates. Additionally, the lower prevalence observed in this study could be attributed to a higher likelihood of animals being exposed to the effects of antibiotics, as many animal owners in the study area administer modern antibiotic drugs and traditional medicine to their animals without a prescription. Frequent use of antibiotics for purposes other than their intended use may also contribute to the lower *Campylobacter* prevalence.

The carriage rate of *Campylobacter* in wastewater, at 9.1%, was higher than that in food or animal feces. This higher prevalence in wastewater is likely attributed to the presence of a mixture of human and animal feces, particularly poultry feces, in the wastewater, leading to an increased occurrence rate. Wastewater serves as an ideal reservoir for pathogens and acts as a common medium for their dissemination [49]. Nonetheless, this finding is lower than the pooled prevalence of 52.97% reported in a meta-analysis of *Campylobacter* in wastewater [50]. Consequently, the utilization of wastewater as irrigation water poses a potential risk of contaminating agricultural produce, particularly vegetables, on farmlands. In sub-Saharan Africa, a significant number of vegetable crops are irrigated with wastewater, providing an opportunity for some microbes to contaminate plants and subsequently affect consumers [51].

Interestingly, *Campylobacter* was not detected in the drinking water, recreation water, or irrigation water samples analyzed in this study. In contrast, a previous study conducted in England identified *Campylobacter* in different water sources [52]. Furthermore, other studies have reported a prevalence of 10% in drinking water from Ethiopia [12] and 52% in groundwater in Nigeria [53] for *Campylobacter*. The detection of *Campylobacter* in water sources is an indication of unsanitary fecal contamination, which can increase the risk of pathogen spread. Nevertheless, it is important to note that the current study only included 40 water samples, so it is difficult to conclude that water sources in eastern Ethiopia are free of *Campylobacter* contamination. Additionally, the scope of this study was restricted to analysing water sources directly used by children or indirectly by the families of the diarrheic children identified during the study period.

The occurrence of *Campylobacter* exhibited clear fluctuations in monthly patterns, peaking in February 2022 (winter month). The second highest occurrence was observed during the rainy season. This is in line with the description that the occurrence of some diarrheagenic FBPs is seasonal, peaking at different times of the year [54]. A review of *Campylobacter* epidemiology in European countries documented that *Campylobacter* cases typically peak during the summer months and have a smaller peak during the winter [41]. Similarly, recent studies conducted in Ethiopia have demonstrated seasonal variations in *Campylobacter* prevalence, with a higher occurrence rate observed during the wet season than during the dry season [43].

The higher occurrence of *Campylobacter* during the winter season in the current study could be linked to water scarcity and the resulting challenges in maintaining proper food sanitation and personal hygiene practices during the dry season, as described in a previous study in Kenya [55]. In many villages in and around the study area, residents often rely on shared water sources (water points) for various purposes, including animal drinking, irrigation, food

preparation, and other domestic uses, particularly when water resources are scarce during the dry season.

The current study identified only *C. jejuni* and *C. coli*, with 62% of the sequenced isolates being *C. jejuni*. This finding is consistent with a recent study conducted in eastern Ethiopia, which exclusively reported these two *Campylobacter* spp.[36]. However, *C. fetus* has also been reported in studies from different parts of Ethiopia [5,12], which could indicate variation in the local epidemiology of *Campylobacter* spp. across different regions and study settings within the country. Nevertheless, the differences in detection methods may also vary the results, as this study employed WGS, which provides a more definitive confirmation. Like the current study, many studies have reported *C. jejuni* as the dominant *Campylobacter* spp. in Ethiopia [5,12,36]. This may be attributed to the unique characteristics of the species. *Campylobacter* spp. are generally considered fastidious microaerophilic pathogens. However, *C. jejuni*, in particular, exhibits a relatively high degree of aerotolerance and starvation survival [56]. Factors such as geographical location, environmental conditions, agricultural practices, food handling and preparation methods, and access to clean water and sanitation can influence the prevalence and distribution of *Campylobacter* spp. within a given region.

The genomic analysis revealed that 69.2% (9 out of the 13) of the sequenced *Campylobacter* case isolates were clustered together with at least one asymptomatic human (caretaker) or environmental source isolate. Although there have been no prior studies that employed WGS analysis in tracing the sources of *Campylobacter* causing diarrhea in children in Ethiopia, the present study underscores the complex transmission dynamics of the pathogen, where it can be shared between human, animal, and environmental reservoirs. Previous source attribution studies have also revealed the involvement of multiple vectors in *Campylobacter* transmission [57,58].

The maximum likelihood phylogenetic analysis also revealed that *Campylobacter* isolates obtained from diarrheic children exhibited three distinct clustering patterns, as indicated by the subclusters (clades) shown in **Fig 3**. In some clades, the case isolates were grouped together with isolates from both asymptomatic human (caretakers) and environmental source exposures, whereas a few clades contained only the case isolates. Additionally, in other clades, the case isolates were clustered exclusively with isolates from different exposure sources without caretaker isolates. Such clustering patterns indicate the presence of multiple routes of *Campylobacter* transmission, which may be resulted from either the genomic or both the genomic and the epidemiological relationships of the isolates [59–61]. The identification of closely related *Campylobacter* isolates from different child cases, despite their temporal separation, also indicates the potential for environmental persistence and transmission of the pathogen [56,62].

In clade G4, a case isolate (11681121) was closely related (SNP < 10) to a *Campylobacter* isolated from a corresponding caretaker in the same household, as well as a bovine meat isolate. Notably, these 3 isolates also shared the same sequence type (ST), a previously unknown ST. According to the caretaker's report, the family, including the diarrheic child, had consumed food prepared with meat purchased from a local market two days prior to the onset of diarrhea. This suggests a transmission link between the household and the food product, with the purchased meat being the probable source of the *Campylobacter* isolated from this diarrheic case. In fact, meat has been widely recognized as a potential source of *Campylobacter* species [57,58].

Similarly, a case isolate (11521121A) and an isolate from poultry (15331354) were clustered in the same clade and closely related (SNP < 10). Clustering of clinical *Campylobacter* isolates with isolates from poultry, indicating their genetic similarities, has also been previously reported in Ireland [61]. When genome sequences cluster together, it is potentially suggestive

of isolates from a common source and can provide support for the detection and investigation of outbreaks [59]. However, this study is constrained to incorporating molecular clock analysis to demonstrate the principal transmission directions (routes) of *Campylobacter* isolates with such clustering patterns.

In clade G2, three closely related *Campylobacter* isolates sharing the ST353 profile were identified from different child cases, even though they were admitted to healthcare facilities weeks to months apart. This suggests that once the first case (1891121A) acquired the pathogen from an unidentified source, the environment became contaminated, which could have served as the source of infection for the subsequent cases. The persistence of *Campylobacter*, particularly *C. jejuni*, in the environment over extended periods and its potential transmission through various vehicles have been previously reported [56,62]. MLST identifies bacterial lineages by indexing variation in seven housekeeping genes, with strains sharing a central genotype(ST) having a common ancestor and often exhibiting shared phenotypic properties, including niche associations and virulence[30,60,61,63].

However, this finding suggests that the primary source of the pathogen for the first case (1891121A) remained unidentified; in fact, some potential exposure sources, such as leftover food, were not available during tracing in many households, as the prepared food was often not stored. Likewise, a study conducted in Tanzania on *Salmonella* source tracing reported the unavailability of traceable contact samples, such as leftover food [64]. A study conducted in Denmark also reported that some human isolates are not attributed to any identified sources [60].

The phylogeny revealed that case isolate 11491121 was related to *Campylobacter* isolates from three exposure sources: abattoir-linked meat, goat feces from the local market, and wastewater connected to hotels and restaurants in the area. However, the case isolate was not within the same clade as the isolates from these environmental sources, demonstrating a few changes, as the collection of the child stool and source exposure samples had time differences. *Campylobacters* have a propensity to take up naked DNA from their environment and integrate it into their genetic material which can result in genomic changes over time [65]. Nevertheless, the close resemblance among the isolates from these three exposure sources (clade G1) suggests that the pathogen is circulating in the environment and disseminating through multiple vehicles, potentially heightening the risk of infection in both humans and animals.

Similarly, *Campylobacter* isolates identified from meat, poultry feces from the local market, and a wastewater sample were clustered together in clade G3. This finding indicates the spread of a *Campylobacter* strain across various environmental sources, including animals, food, and wastewater, which probably increases the occurrence of campylobacteriosis. Without human involvement, *Campylobacter* spp. can persist and survive within a relatively hostile environment for up to 10 months [66]. Hence, the current findings further highlight the need to address zoonotic *Campylobacter* transmission from animals and enforce rigorous food safety practices in this locality.

## Conclusion and recommendations

*Campylobacter* is a significant zoonotic foodborne pathogen that poses a major public health threat, especially in low-income nations. This study identified *Campylobacter* in various exposure sources of diarrheic children, including human, animal, food, and environmental samples. Notably, the occurrence of *Campylobacter* did not differ significantly between rural and urban settings, demonstrating a widespread risk of exposure, particularly during the dry season. Whole genome sequencing-based phylogenetic and MLST analyses revealed that most case isolates were closely related, clustering together and sharing sequence types (STs) with isolates from diverse exposure sources. Sequence analysis identified only *C. jejuni* and *C. coli*,

with 8 distinct sequence types (STs), indicating a low *Campylobacter* spp. diversity in eastern Ethiopia, but disseminated across different sources, suggesting the involvement of multiple transmission vectors, including humans, animals, and the environment.

In addition to this complex transmission pathway evidence, regression analysis underscores the need for improved food safety, wastewater management, and handling of animal excreta, especially poultry droppings, to reduce environmental spread and *Campylobacter*-associated childhood diarrhea in both Harar town and Kersa district. This study demonstrates the importance of a one health approach using WGS to trace the sources of *Campylobacter* in diarrheic children, which may also apply to other zoonotic pathogens. However, more in-depth analyses, such as molecular clock and source attribution models, are further needed to determine specific transmission routes (directions) and the relative contributions of each source.

## Acknowledgments

The authors would like to thank FOCAL research project funders and the Haramaya University School of Medical Laboratory for their cooperation in conducting the activities in their laboratory. We are also grateful to the FOCAL research project data collection and laboratory analysis teams.

## Author Contributions

**Conceptualization:** Dinaol Belina, Tesfaye Gobena, Tine Hald.

**Data curation:** Dinaol Belina, Cecilie Amalie Neijendam Thystrup.

**Formal analysis:** Dinaol Belina, Cecilie Amalie Neijendam Thystrup.

**Funding acquisition:** Tesfaye Gobena, Tine Hald.

**Investigation:** Dinaol Belina.

**Methodology:** Dinaol Belina, Tesfaye Gobena.

**Project administration:** Tesfaye Gobena, Tine Hald.

**Resources:** Tesfaye Gobena, Bahar Mummed.

**Software:** Cecilie Amalie Neijendam Thystrup, Tine Hald.

**Supervision:** Tesfaye Gobena, Ameha Kebede, Meseret Chimdessa, Tine Hald.

**Validation:** Tine Hald.

**Writing – original draft:** Dinaol Belina.

**Writing – review & editing:** Dinaol Belina, Tesfaye Gobena, Ameha Kebede, Meseret Chimdessa.

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
