## [Decision Letter · Decision Letter 0]

31 Jul 2024

PGPH-D-24-01096

Detection and Occurrence Patterns of Campylobacter species in the Exposure Environments of Diarrheic Children in Ethiopia

Dear Dr. Belina,

Thank you for submitting your manuscript to PLOS Global Public Health. After careful consideration, we feel that it has merit but does not fully meet PLOS Global Public Health’s publication criteria as it currently stands. Therefore, we invite you to submit a revised version of the manuscript that addresses the points raised during the review process.

We look forward to receiving your revised manuscript.

Kind regards,

Mohan Amarasiri

Academic Editor

Journal Requirements:

1. In the ethics statement in the Methods, you have specified that verbal consent was obtained. Please provide additional details regarding how this consent was documented and witnessed, and state whether this was approved by the IRB

a. Please clarify all sources of funding (financial or material support) for your study. List the grants (with grant number) or organizations (with url) that supported your study, including funding received from your institution. 

3. We do not publish any copyright or trademark symbols that usually accompany proprietary names, eg (R), (C), or TM  (e.g. next to drug or reagent names). Please remove all instances of trademark/copyright symbols throughout the text, including TM on page 11.

4. In the online submission form, you indicated that "The data used in this study are presented in the document, and complete data, including supplementary files (if any), may be accessible through the corresponding author." 

3. Uploaded as supplementary information.

Additional Editor Comments (if provided):

Reviewers' comments:

Reviewer's Responses to Questions

**Comments to the Author**

1. Does this manuscript meet PLOS Global Public Health’s publication criteria? Is the manuscript technically sound, and do the data support the conclusions? The manuscript must describe methodologically and ethically rigorous research with conclusions that are appropriately drawn based on the data presented.

Reviewer #1: Yes

Reviewer #2: Yes

2. Has the statistical analysis been performed appropriately and rigorously?

Reviewer #1: Yes

Reviewer #2: Yes

3. Have the authors made all data underlying the findings in their manuscript fully available (please refer to the Data Availability Statement at the start of the manuscript PDF file)?

Reviewer #1: Yes

Reviewer #2: Yes

4. Is the manuscript presented in an intelligible fashion and written in standard English?

Reviewer #1: Yes

Reviewer #2: Yes

5. Review Comments to the Author

Reviewer #1: General comments

The work is presented in a journal style, the methodology is clear, straightforward and acceptable for this kind of a work, Paper should be accepted with some minor corrections.

General comments are to avoid personifications in the writing, too long sentences and check for some few grammatical errors in the paper.

Abstract

Maintain using single decimal or no decimal place (see line 21)

No results from the genomic and bioinformatics parts of the method

Introduction

Looks like the study is done on animals, food and environmental samples in immediate diarrheic environment of UFC and that is not captured in the tittle.

Need to be clearer in the problem statement and justification, develop a clear foundation as to why you had to do the study.

Re-construct the objective to be simple and clear to the reader

Methods

Based on health indicators, what was the cue to go to into that specific study site? This information on is not captured

Be aware of the use of the word tracing and tracking, in a context of the paper its more of tracing rather than tracking.

How was whole genome sequencing done, what method was used, was it sanger, next generation? No enough details.

Detailed explanation of bioinformatics analysis is missing

Results

There is a consistent omission of the whole genome species identification data, what was the output of sequence data? similarity index of the species to the Reference in GeneBank? No any plot from sequencing?

Consistence with the use of decimal places, choose to either use one or two or no decimal places in proportions presentations,

Discussion

The discussion is well written, it covers all the important parts of the results, however, there has to be a link between the identification used and its influence on the results of identification.

Reviewer #2: The study on the Detection and Occurrence Patterns of Campylobacter species in the Exposure Environments of Diarrheic Children in Ethiopia is well grounded, well thought-out and scientifically highly revealing with clear exposition of some risk factors that contribute to the establishment and propagation of Campylobacter spp., the species types and prevalence in one urban and one rural settings in Ethiopia. The script is simple to understand and the discussions are well laid out with smooth flow of information. The authors may wish to recast the sentence in line 324 which states ‘’However, (10) reported 9.3% Campylobacter spp. prevalence in food in Ethiopia’’ to, e.g. However, one other study (10) has reported 9.3% Campylobacter spp. prevalence in food in Ethiopia. The same also goes for lines 383 to 385 ‘’ The overall prevalence of Campylobacter in animals in this study is lower than the findings of previous studies conducted by (12) and (10), who, respectively, reported 56.5% and 14.6% which may read:

The overall prevalence of Campylobacter in animals in this study is lower than the 56.5% reported for Lare District (12) and 14.6% for Jima town (10) of Ethiopia.

6. PLOS authors have the option to publish the peer review history of their article (what does this mean?). If published, this will include your full peer review and any attached files.

**Do you want your identity to be public for this peer review?** For information about this choice, including consent withdrawal, please see our Privacy Policy.

Reviewer #1: **Yes: **Agnes A. Mpinga

Reviewer #2: No

---

## [Decision Letter · Decision Letter 1]

24 Sep 2024

PGPH-D-24-01096R1

Occurrence and Diversity of Campylobacter species in Diarrheic Children and Their Exposure Environments in Ethiopia

Dear Dr. Belina,

Thank you for submitting your manuscript to PLOS Global Public Health. After careful consideration, we feel that it has merit but does not fully meet PLOS Global Public Health’s publication criteria as it currently stands. Therefore, we invite you to submit a revised version of the manuscript that addresses the points raised during the review process.

We look forward to receiving your revised manuscript.

Kind regards,

Mohan Amarasiri

Academic Editor

Journal Requirements:

1. In the ethics statement in the Methods, you have specified that verbal consent was obtained. Please provide additional details regarding how this consent was documented and witnessed, and state whether this was approved by the IRB.

Additional Editor Comments (if provided):

Reviewers' comments:

Reviewer's Responses to Questions

**Comments to the Author**

1. If the authors have adequately addressed your comments raised in a previous round of review and you feel that this manuscript is now acceptable for publication, you may indicate that here to bypass the “Comments to the Author” section, enter your conflict of interest statement in the “Confidential to Editor” section, and submit your "Accept" recommendation.

Reviewer #2: (No Response)

Reviewer #3: All comments have been addressed

2. Does this manuscript meet PLOS Global Public Health’s publication criteria? Is the manuscript technically sound, and do the data support the conclusions? The manuscript must describe methodologically and ethically rigorous research with conclusions that are appropriately drawn based on the data presented.

Reviewer #2: Yes

Reviewer #3: Yes

3. Has the statistical analysis been performed appropriately and rigorously?

Reviewer #2: Yes

Reviewer #3: Yes

4. Have the authors made all data underlying the findings in their manuscript fully available (please refer to the Data Availability Statement at the start of the manuscript PDF file)?

Reviewer #2: Yes

Reviewer #3: Yes

5. Is the manuscript presented in an intelligible fashion and written in standard English?

Reviewer #2: Yes

Reviewer #3: Yes

6. Review Comments to the Author

Reviewer #2: The manuscript centred on detection of Campylobacter species in under fives with acute diarrhoea and testing for the organism among the family (caretaker) contacts of the child as well as the environment where the diarrheic cases are found including remnant of foods etc. The project takes in so many areas of investigation and tried to establish presence of two major campylobacter species.

The manuscript looks quite a bit voluminous. It will be good if the introduction and discussion could be reduced by discussing more of the results and moderating comparisons with similar works.

Lines 93: remove sample(s)

Line 102: for the recent exposure details (let’s define the recent e.g was it 2 or 3 days or more?)

Line 134: Sentence should be closed with a bracket. What measures did you put in place to ensure that the stool samples handed over to you by the caretakers of the child really belonged to them?

Line 211: A little confusion here. You mean the child’s stool was collected at health care facilities or are remnants of the stool collected at the health care facility for testing prior to treatment.

Line 234: Please decide on whether to use one decimal place or two to make the figures uniform.

Line 266: Please let the audience know what constituted the exposure environment from which the graph was derived. Is it the same thing as items on lines 118-120?

Line 324: Please recast the statement from However (10) reported 9.3% Campylobacter species..........to e.g. Other workers reported 9.3% Campylobacter species involvement ……………… (10)

Lines 384-385: Please change to …… of previous studies conducted by other workers which gave 56.5%f (12) and 14.6% (10).

Line 416: may lead to an enhancement?? In what way please??

Reviewer #3: I think authors not necessory to show the pictures of the The spectrophotometric readings for DNA purity obtained from some samples. I recomonded removing those and just mentiond the DNA concentration.

7. PLOS authors have the option to publish the peer review history of their article (what does this mean?). If published, this will include your full peer review and any attached files.

**Do you want your identity to be public for this peer review?** For information about this choice, including consent withdrawal, please see our Privacy Policy.

Reviewer #2: No

Reviewer #3: No

---

## [Editor Report · Decision Letter 2]

10 Oct 2024

Occurrence and Diversity of Campylobacter species in Diarrheic Children and Their Exposure Environments in Ethiopia

PGPH-D-24-01096R2

Dear Dr Belina,

We are pleased to inform you that your manuscript 'Occurrence and Diversity of Campylobacter species in Diarrheic Children and Their Exposure Environments in Ethiopia' has been provisionally accepted for publication in PLOS Global Public Health.

Best regards,

Mohan Amarasiri

Academic Editor